# Micro-Computed Tomography as a complementary tool for histopathological diagnosis of oral soft tissue lesions – Proof of concept

Lazar Kats[1], Yankel Gabet[2], Uzi Shpunt[1], Marilena Vered [1,3]*

1 Department of Oral Pathology, Oral Medicine and Maxillofacial Imaging, Goldschleger School of Dental Medicine, Gray Faculty of Medical and Health Sciences, Tel Aviv University, Tel Aviv, Israel, 2 Department of Anatomy and Anthropology, Gray Faculty of Medical and Health Sciences, Tel Aviv University, Tel Aviv, Israel, 3 Institute of Pathology, The Chaim Sheba Medical Center, Tel Hashomer, Ramat Gan, Israel

* mvered@tauex.tau.ac.il

## Abstract

### Background

Accurate diagnosis of oral soft tissue lesions is critical for effective treatment, yet conventional histopathological examination, the gold standard, faces limitations. These include two-dimensional (2D) visualization and malorientation, which can obscure critical diagnostic features, like epithelial-connective tissue interfaces. Micro-computed tomography (µCT) offers a non-destructive, high-resolution three-dimensional (3D) imaging alternative to address these challenges. Still, its use for soft tissue visualization is limited. We tested a method with specific radio-opaque staining and µCT scanning settings to visualize oral soft tissue biopsies as a proof of concept.

### Methods

Biopsies from 12 patients with different oral mucosa lesions were stained with Lugol's iodine, scanned at 3µm resolution with 70kV energy, and the resulting volumes were compared to histopathological sections by specialists in oral radiology and oral pathology.

### Results

µCT produced 2D images with tissue architecture comparable to hematoxylin and eosin (H&E)-stained sections, distinguishing epithelium, connective tissue, and keratin, while 3D reconstructions revealed topographic details, such as ulceration depth and vascular patterns, unattainable in histopathology.

**Data availability statement:** All relevant data are within the paper.

**Funding:** Purchase of the µCT was supported by Tel Aviv University starter funds and by Israel Science Foundation (ISF) grant 1822/12 to YG.

**Competing interests:** NO authors have competing interests.

## Conclusions

These findings highlight µCT potential as a complementary diagnostic tool, enhancing topographic rendering while preserving tissue integrity. Standardized protocols and broader validation, particularly for precancerous and malignant lesions, are essential for clinical adoption, promising improved diagnostic accuracy in oral pathology.

## Introduction

The importance of accurate diagnosis cannot be overstated in medicine and dentistry, where it serves as the cornerstone of effective patient treatment, directly influencing therapeutic decisions and outcomes [1]. In the oral cavity, soft tissue lesions present a diverse array of pathologies, ranging from benign conditions like irritation fibroma and frictional keratosis to complex immune-mediated disorders like oral lichen planus, and malignancies such as oral squamous cell carcinoma (OSCC), which constitutes over 90% of oral cancers [2,3]. These lesions challenge clinicians due to their histological heterogeneity, subtle clinical presentations, and potential for malignant transformation, particularly in precancerous conditions like leukoplakia or erythroplakia [4,5]. Despite rapid advancements in medical technology, histopathological examination has remained largely unchanged for decades and continues to be the gold standard for diagnosing oral lesions [6]. However, reports dating back over 20 years have highlighted its problematic nature, prompting exploration of complementary techniques to enhance diagnostic precision [7].

Histopathological examination involves a multi-step process. It starts with biopsy collection, fixation, embedding, sectioning, and staining (typically with hematoxylin and eosin, H&E). The slides are then examined under a microscope by a trained pathologist who interprets characteristic diagnostic features based on extensive training [8]. Pathologists combine these findings with clinical context to render a diagnosis, a process that, while robust, is susceptible to errors at various stages—from biopsy collection to laboratory processing and final interpretation [9,10]. One of the most critical technical errors is malorientation of tissue sections, particularly in soft tissue biopsies where precancerous or malignant changes in the lining epithelium are suspected [11]. Maloriented sections may fail to display the full epithelial layering or its relationship to underlying connective tissue, both essential for accurate diagnosis. For example, excluding the basal epithelial layer can obscure signs of dysplasia, leading to misdiagnosis of benign versus malignant pathology [12]. Additionally, histopathology's fundamental limitation lies in its two-dimensional (2D) representation of tissue, which cannot fully capture the three-dimensional (3D) architecture critical for understanding lesion topography [13]. Inter-observer variability further complicates accuracy, with discordant diagnoses reported in 10–30% of oral lesions of the oral potentially malignant disorder types, particularly those addressing dysplasia grading [14,15].

Given these limitations, there is a pressing need for an adjunctive method that can penetrate the entire tissue volume, preserve its topographic structure, and present it

in 3D, thereby enhancing the accuracy of histopathological diagnosis [16]. The oral cavity unique environment, marked by exposure to carcinogens (e.g., tobacco, alcohol), chronic inflammation, and a global OSCC incidence exceeding 350,000 cases annually, calls attention to this need, particularly in low- and middle-income countries where diagnostic resources are scarce [3,17]. Micro-computed tomography (μCT) emerges as a promising candidate, offering non-destructive, high-resolution 3D imaging that addresses histopathology shortcomings [18].

μCT, an advanced derivative of conventional computed tomography (CT), achieves resolutions in the micron range (1–5 μm), far surpassing CT millimeter-scale capabilities, making it ideal for small specimens like oral biopsies [19]. Unlike CT, which is used for gross anatomical imaging, μCT allows detailed visualization of tissue microstructure without physical sectioning, preserving sample integrity [20]. A key advantage is its ability to generate multi-plane images (axial, coronal, sagittal) and 3D reconstructions, overcoming histopathology dependence on slice orientation and 2D limitations [21]. The non-destructive nature of μCT enables repeated scans, digital storage of imaging data (including 2D/3D images and videos), and easy sharing across laboratories, enhancing collaborative research and diagnostic review [22]. Furthermore, μCT offers relatively short acquisition times compared to other high-resolution imaging modalities, making it a practical tool for research and potential clinical applications.

Initially developed for industrial applications, such as material defect analysis, μCT gained prominence in biomedical research for studying calcified tissues like bone and teeth [23,24]. In dentistry, it has revolutionized the analysis of enamel defects, caries progression, periodontal bone loss, and implant osseointegration, providing quantitative volumetric data unattainable with 2D radiography [25]. Recent advancements have extended μCT's utility to soft tissues, particularly in small animal models, overcoming the challenge of low X-ray absorption through contrast-enhancing agents [26]. Early studies used osmium tetroxide, prized for its ability to highlight diverse soft tissues (e.g., cardiovascular systems), but its toxicity and complex preparation limited its practicality [27]. Lugol's iodine, a simpler and less toxic alternative, has since become standard, enabling high-contrast 3D imaging of soft tissues, such as mouse embryos, where it delineated vasculature, neural networks, and organ systems with exceptional clarity [28,29]. These scans revealed critical developmental defects, like cardiac anomalies, often missed by conventional histopathology, highlighting μCT diagnostic potential [30].

Since 2010, innovations in contrast agents (e.g., phosphotungstic acid [PTA], nanoparticle-based compounds) and hardware (e.g., photon-counting detectors, high-flux X-ray sources) have pushed μCT's resolution to sub-micrometer levels and reduced scanning times, broadening its applications [31]. μCT's non-destructive nature also allows integration with other techniques, such as microscopy or immunohistochemistry, and supports the analysis of rare or irreplaceable specimens, like museum samples or archival biopsies, without compromising their integrity [32]. Despite these advances, no studies prior to 2020 systematically investigated μCT for human oral soft tissues, as reviews of the field primarily focused on calcified tissues or non-oral soft tissues [18].

The aim of this study was to serve as a proof of concept regarding the ability of μCT to visualize soft tissue biopsies from oral cavity lesions requiring diagnosis and treatment, develop appropriate protocols for staining and scanning, and compare the results with routine histopathological sections. The working hypothesis is that μCT will provide high-quality images with good contrast in multiple planes, and 3D reconstructions, albeit with less cellular detail compared to histopathology. By addressing histopathology's technical limitations, μCT could enhance diagnostic accuracy, offering a complementary approach to improve patient care in oral pathology.

## Materials and Methods

The study was approved by the Ethics Committee of Tel Aviv University, 1494–5/2022. Patients that participated in the study were referred from community dental surgeons to the Oral Medicine Clinic, School of Dental Medicine, Tel Aviv University, due to identification of oral mucosal lesions for purposes of diagnosis and treatment; they were not recruited for the specific purpose of this study. Since our clinic is a teaching platform for both under-graduate and post-graduate students, during registration procedures and a priori to entering the dental office, all patients are routinely presented with



a form of written informed consent, in which they can choose to agree that any information from their files can be used as teaching/publishing material in an anonymized way. Patients that refute to share information on their health status, diagnosis, treatment and outcomes are being taken care of equally.

Soft tissue biopsies were collected between April 3rd, 2022 – February 28th, 2023 from 12 patients (7 females, 5 males; age range 30–77 years). Inclusion criteria conditioned that patients have not been given any local or systemic medications for their oral lesions, nor have undergone any previous surgical procedure in the lesions' area. Clinically, selected lesions were categorized either as those with white/white-red color change of the oral lining epithelium, or those that looked exophytic and reactive. The attempted submitted clinical diagnoses are detailed in Table 1. In addition, included were lesions of an adequate size that could enable macroscopic sectioning into halves, so that at least an initial microscopic diagnosis could be submitted with no delay, while the other half was being prepared for µCT scanning. Lesions clinically suspected for malignancy were excluded from study.

Each biopsy was divided into two halves. One half was fixed in 10% formalin for 24 hours, processed routinely, and stained with H&E for histopathological examination. The other half was fixed similarly, washed with tap water, and stained with 1% iodine and 2% potassium iodide (Lugol's solution) for 12 hours [32]. Post-scanning, this half was washed and processed for H&E staining to confirm compatibility with histopathology. To mention that final diagnostic reports were sent to the clinicians immediately after the scanned part of the biopsy has been H&E-stained. No diagnosis has been based on the scans, as these were intended only for research and not diagnostic purposes. None of the submitted microscopic diagnoses posed diagnostic challenge that could have impacted the clinicians' decision-making regarding treatment, but rather served as an architectural/morphological base-line for comparison with the corresponding scanned images. Actual numbers of the biopsies as well as their corresponding scanned volumes were coded with the same random number. Only the oral pathologist (MV) had the key to the codes in order to identify the biopsy number of the scanned tissue so as to complete the pathology report. After submission of all final reports, codes were permanently deleted.

Scans were performed using a µCT50 scanner (Scanco Medical AG, Switzerland) at the Department of Anatomy & Anthropology, Faculty of Medical and Health Sciences. The protocol used 3 µm isotropic nominal resolution, 70 kV energy, 200 µA intensity, 1000 projections, and 1500 msec integration time.

**Table 1. Patient demographics and lesion characteristics.**

| Patient No. | Age | Sex | Location | Macroscopic dimensions (cm) | Clinical Diagnosis | Biopsy Type | Histopathological Diagnosis |
|---|---|---|---|---|---|---|---|
| 1 | 62 | F | Lower lip mucosa | 0.6 × 0.5 × 0.3 | Irritation fibroma | Excision | Irritation Fibroma |
| 2 | 57 | M | Buccal mucosa | 0.5 × 0.5 × 0.2 | Frictional keratosis | Incision | Acanthosis, Hyperkeratosis |
| 3 | 30 | M | Dorsal tongue | 0.4 × 0.3 × 0.3 | Irritation fibroma | Excision | Giant Cell Fibroma |
| 4 | 56 | F | Buccal mucosa | 0.5 × 0.4 × 0.3 | OLP | Incision | Oral Lichen Planus |
| 5 | 32 | F | Buccal mucosa | 0.4 × 0.6 × 0.9 | Irritation fibroma | Excision | Irritation Fibroma |
| 6 | 58 | M | Buccal mucosa | 0.7 × 0.5 × 0.3 | Irritation fibroma | Excision | Irritation Fibroma |
| 7 | 36 | F | Gingival papilla | 1.0 × 1.0 × 1.5 | Pyogenic granuloma | Excision | Pyogenic Granuloma |
| 8 | 31 | M | Hard palate | 1.0 × 0.6 × 0.4 | HPV-related lesion | Excision | Fibro-Epithelial Hyperplasia |
| 9 | 65 | F | Gingival papilla, tooth 42 | 0.8 × 0.7 × 0.4 | Pyogenic granuloma | Excision | Fibrous Epulis, ulcerated |
| 10 | 77 | F | Gingiva around implant 46 | 0.4 × 0.8 × 0.9 | Pyogenic granuloma | Excision | Pyogenic Granuloma |
| 11 | 67 | F | Lateral tongue | 0.7 × 0.4 × 0.3 | Fibro-epithelial hyperplasia | Excision | Fibro-Epithelial Hyperplasia |
| 12 | 69 | M | Tongue tip | 1.0 × 0.6 × 0.2 | Irritation fibroma | Excision | Fibro-Epithelial Hyperplasia with koilocytosis |

F: Female, M: Male; HPV – human papilloma virus.

µCT 2D and 3D reconstructions were analyzed by two experts in oral medicine and radiology. They focused on identification of the tissue architecture (epithelium, connective tissue, blood vessels) in parallel to the H&E-stained sections, which were evaluated by an oral pathologist. Results were compared qualitatively for diagnostic concordance. Scanco proprietary software (Scanco Medical AG, Switzerland) and RadiAnt Dicom Viewer (v. 5.5, Medixant, Poland), were used for 2D and 3D volume analyses.

## Statistical Analysis

Qualitative comparisons assessed tissue visualization, contrast, and topographic features between µCT and histopathology. No quantitative statistical tests were applied due to the study's exploratory nature.

## Results

### Patient and Lesion Characteristics

Table 1 summarizes the demographics, lesion locations, clinical diagnoses, biopsy types, and histopathological findings for the patients. Twelve participants, 7 females and 5 males, aged 30–77 years, participated in the study. Incisional or excisional biopsies were obtained from the lower lip, cheek mucosa, tongue, palate, and gums. Lesions ranged from 0.4 × 0.3 × 0.2 cm to 1.0 × 1.0 × 1.5 cm, with clinical and histopathological diagnoses including irritation fibroma (n = 3), fibro-epithelial hyperplasia (n = 3), pyogenic granuloma (n = 2), fibrous epulis (n = 1), giant cell fibroma (n = 1), oral lichen planus (n = 1), and acanthosis with hyperkeratosis (n = 1).

### Case Studies

**Case 1 – Giant Cell Fibroma (Patient 3 in Table 1).** The standard H&E-stained histological section shows a lesion that protrudes above the tongue surface and is filled with connective tissue. It should be noted that the tissue shows no alterations or artefacts following Lugol immersion or potential heating effects from the µCT high resolution imaging procedure. The lesion measures approximately 3 mm in width and 4 mm in height. The two-dimensional tomographs from this half of the biopsy were selected to best fit the H&E-stained section presented in the microscopic plane. Since the paraffin block undergoes a certain degree of trimming in order to achieve the best quality tissue section, the microscopic section is very similar, albeit not identical, to its corresponding tomograph. Hence, this case (as well as in the following cases), the two-dimension tomographs clearly demonstrate a tissue architecture consistent with the histological section. The Lugol solution stained the different layers of tissue in a way that clearly discriminated between the keratin (highest radioopacity), the epithelium, and the connective tissue (least radiopaque). These are well defined and can be depicted in virtually any plan. Blood vessels are also clearly visible within the connective tissue with a radioopacity similar to that of the epithelium. The µCT tomographs can also be displayed as entire volumes (stacked sections) for a better appreciation of the lesion from different angles, which is neither accessible nor practical in histological sections. The surface epithelium surrounding the lesion appears intact with a topography of small protrusions and an ulcer is clearly visible as a discontinuation of the epithelium on one aspect – these findings were not observed in the H&E-stained sections.

**Case 2 – Acanthosis and hyperkeratosis (Patient 2 in Table 1).** The standard H&E-stained histological section shows a lesion with thickened lining epithelium (acanthosis) and hyperkeratosis – features that are abnormal at this location. As mentioned for Case 1, the tissue shows no artefacts following scanning procedure. In addition, similar to Case 1, the Lugol solution stained the epithelium and the underlying connective tissue differently and this was reflected in the scans as different intensities of radioopacity. Bundles of striated muscles was an additional tissue component that was highlighted in the scans at a high resolution, so that their network of spatial arrangement became visible and could be assessed from different planes.

**Case 3 – Fibrous Epulis, ulcerated (Patient 9 in** Table 1**).** The principle of similarity between the H&E-stained and the corresponding scanned 3D plane is readily apparent, as in the previously described cases. In addition, an ulcerated area that was visible in the H&E-stained section could have been missed in case the specimen had been sectioned in deeper levels, whereas the ulcerated area cannot be missed by the 3D μCT scans. Similarly to the former cases, tissue section is devoid of any artefacts following its scanning.

## Discussion

This study systematically investigated μ-computed tomography (μCT) for visualizing oral soft tissue biopsies. By offering a non-destructive, high-resolution three-dimensional (3D) imaging, this approach addresses the limitations of conventional histopathological examination [13]. Our findings confirm that μCT, using Lugol staining, produces two-dimensional (2D) images with tissue architecture comparable to hematoxylin and eosin (H&E) sections and 3D reconstructions that reveal topographic details unattainable with routine histopathology, supporting our hypothesis that μCT provides high-contrast, multi-planar visualization albeit with less cellular detail [28].

μCT's non-destructive imaging is a transformative advantage over histopathology, which is susceptible to artifacts from tissue manipulation, including shrinkage, tearing, or malorientation [11]. By preserving biopsy integrity, μCT enables subsequent analyses, such as immunohistochemistry, molecular profiling, or repeated imaging, enhancing diagnostic and research workflows [33]. In Case 1 (Giant Cell Fibroma, Fig. 1), 2D images were able to provide details on tissue structures (blood vessels) with a resolution of ~20μ with no pixelation effect, and 3D reconstructions highlighted an ulcer and epithelial projections not fully captured in H&E sections, demonstrating μCT's ability to provide topographic context critical for protrusive or ulcerated lesions [5]. Similarly, Case 3 (Fibrous Epulis, Fig. 3) revealed ulceration topography, a key indicator of lesion severity often obscured in 2D sections due to orientation errors [2]. In Case 2 (Acanthosis and Hyperkeratosis, Fig. 2), μCT's 2D slices aligned with H&E findings, clearly delineating thickened epithelium and hyperkeratosis, while 3D reconstructions revealed a smooth epithelial surface. The visualization of blood vessels in Case 1 (Fig. 1b) suggests potential for vascular lesions like pyogenic granuloma, mirroring μCT's success in delineating embryonic vasculature in animal models [29]. The oral cavity's histological complexity—thin epithelium, dense connective tissue, and high vascularity—exacerbates histopathology's challenges, particularly malorientation, which can obscure epithelial-connective tissue interfaces critical for diagnosing conditions like leukoplakia or lichen planus [7]. μCT's multi-planar imaging mitigates this by enabling virtual sectioning in any orientation. By providing a comprehensive view of lesion architecture, μCT enhances diagnostic confidence, particularly for heterogeneous lesions where sampling errors are common [34].

μCT's ability to address malorientation errors tackles a critical flaw in histopathology, where improper section alignment can obscure diagnostic features, such as basal layer integrity in lichen planus or invasion patterns in early OSCC [12]. For benign lesions like irritation fibroma, μCT clarifies structural relationships, aiding conservative management decisions, such as monitoring versus excision. For precancerous lesions, its visualization of tissue interfaces and blood vessels could enhance early detection, crucial given OSCC's 50–60% five-year survival rate [3]. In research, μCT's preservation of rare or archival specimens enables repeated scans to study lesion progression, treatment effects, or histological correlations, a capability demonstrated in embryology and paleontology [29,32]. For example, μCT could analyze biopsy archives to correlate 3D structural features with clinical outcomes, advancing personalized medicine.

The predominance of benign lesions in our cohort reflects common clinical presentations but limits insights into malignancy, where diagnostic stakes are higher [15]. Future studies should also include precancerous lesions (e.g., leukoplakia) or OSCC to align μCT with global oral cancer priorities, given OSCC's rising incidence (over 350,000 cases annually) and high mortality in low-resource settings [3]. μCT's 3D data could guide surgical planning by mapping lesion boundaries, potentially improving margins and reducing recurrence. It could also inform biopsy site selection by identifying representative areas in heterogeneous lesions, minimizing sampling errors—a frequent issue in oral pathology [34]. Furthermore,

µCT's digital output supports telepathology, similar to standard light microscopy, enabling remote consultation with experts, which is vital in underserved regions [35].

Recent literature highlights µCT's potential in oncology, particularly for detecting malignant lesions, though studies remain preliminary and focused on non-oral soft tissues [26,36]. For example, a 2024 study by Laguna-Castro et al. used eosin-stained µCT to visualize metastatic foci in axillary lymph node biopsies from breast cancer patients, demonstrating improved topographic clarity over 2D histopathology [36]. Yu et al. used µCT to assess depth of invasion in OSCC and found a strong correlation between the histological, hematoxylin and eosin-stained sections and the µCT measurements, in addition to its ability to distinguish various tissue structures, such as tumor tissue, epithelial tissue, muscle tissue and blood vessels [37]. Our study broadens this scope by demonstrating µCT's efficacy across benign, reactive, and HPV-related lesions, highlighting its versatility. Compared to Cone-Beam CT (CBCT), which achieves 100–200 µm resolution and is optimized for in vivo dental imaging, greatest µCT's 3 µm resolution excels for ex vivo biopsy analysis, offering superior structural detail [38]. Magnetic resonance imaging provides soft tissue contrast but lacks µCT's microstructural precision, making it less suitable for small specimens. Intraoral ultrasound enables real-time imaging but cannot match µCT's 3D topographic clarity, particularly for subsurface architecture. Optical coherence tomography (OCT), capable of ~10 µm resolution, is limited to superficial layers (~2 mm depth), whereas µCT images entire biopsies, capturing deeper structures [39]. Confocal microscopy offers cellular detail but lacks µCT's volumetric imaging, restricting its topographic

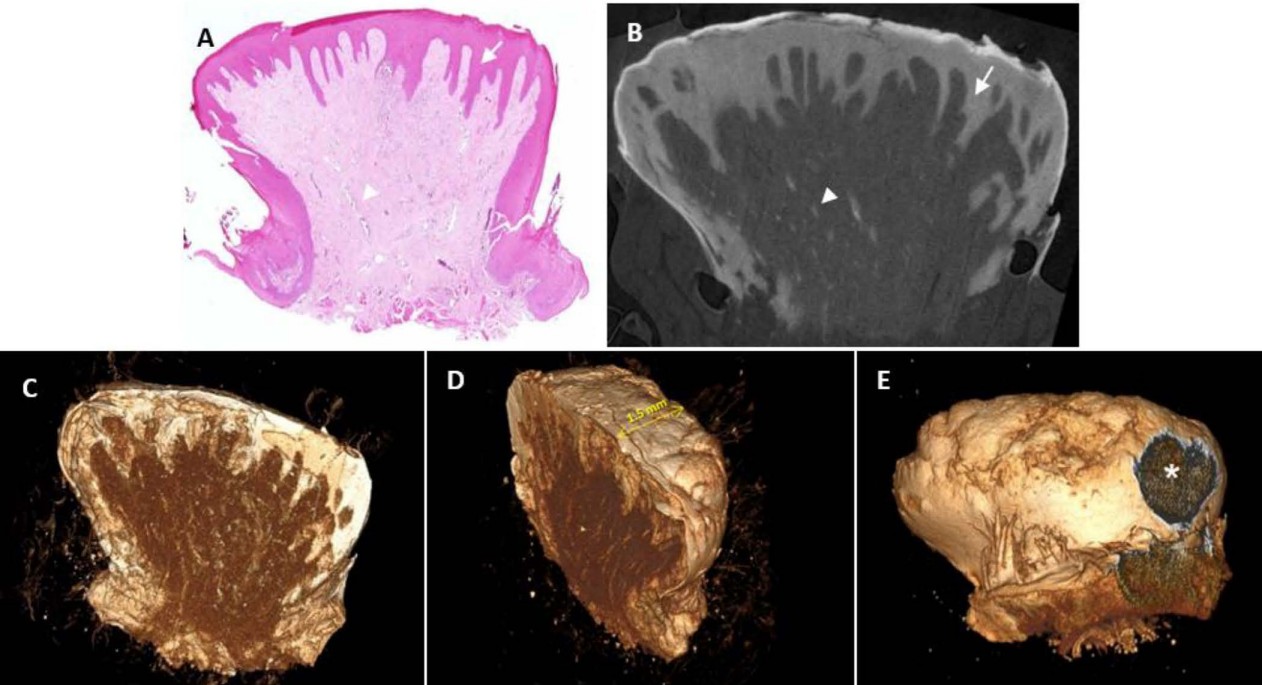

**Fig 1. Giant cell fibroma. A.** Histological section stained with hematoxylin and eosin. The arrow indicates the epithelial lining. The arrow head indicates blood vessels of an assessed diameter of 20-25µ. **B.** Two-dimensional µCT section from the second half of the same biopsy as in A, demonstrating different radioopacity levels across the different layers, i.e., keratin (the thin band coating the lesion, appearing white), the epithelium (arrow), and the connective tissue (least radiopaque). Blood vessels (arrowhead) are clearly visible within the connective tissue, within a diameter range of ~20µ with no pixelation effect, similar to those seen in **A. C–E.** Three-dimensional volume of the entire half biopsy from various angles using pseudocolors that match the tomographic radioopacity (non-segmented). Scale bar: see in D, the tissue thickness is approximately 1.5 mm. The asterisk indicates a small area of epithelial discontinuity – an ulcer **(E)**.



**Fig 2. Acanthosis and Hyperkeratosis. A.** Histopathological section stained with H&E. The epithelial lining appears slightly thickened (acanthosis), with keratin slightly thicker than typical for buccal mucosa, a band of connective tissue, and clusters of striated muscle. The lesion measures approximately 3 mm in width and 3 mm in height. **B.** Two-dimensional µCT section demonstrating findings consistent with the histopathological section. **C–D.** Three-dimensional reconstruction sections showing the lesion from various angles, which are neither accessible nor practical in histological sections. The surface epithelium (white arrow, C) is clearly visible with a relatively smooth and regular topography. Bundles of striated muscles and their spatial arrangement with intersecting fibers in different directions are highlighted in these Lugol-stained sections (black arrows).

insights [40]. Histopathology remains unmatched for cellular resolution, essential for grading dysplasia or confirming malignancy, positioning µCT as a complementary tool [6].

In the present study, Lugol's iodine staining effectively enhanced contrast and provided anatomical information close to histological resolution, distinguishing epithelium (gray), connective tissue (dark gray), and keratin (white), consistent with prior soft tissue studies [28], however it was unable to provide details at a cellular level. The 12-hour staining duration poses a significant barrier to clinical adoption, as rapid diagnostics are critical for suspected malignancies, where delays can worsen prognosis [41]. In addition, if concentration or staining duration with Lugol's iodine is not optimally controlled, there is risk of contrast-induced artefacts following tissue shrinkage or hardening. Furthermore, lack of standardized protocols with no universally accepted staining and acquisition protocol for soft tissues, could affect reproducibility between

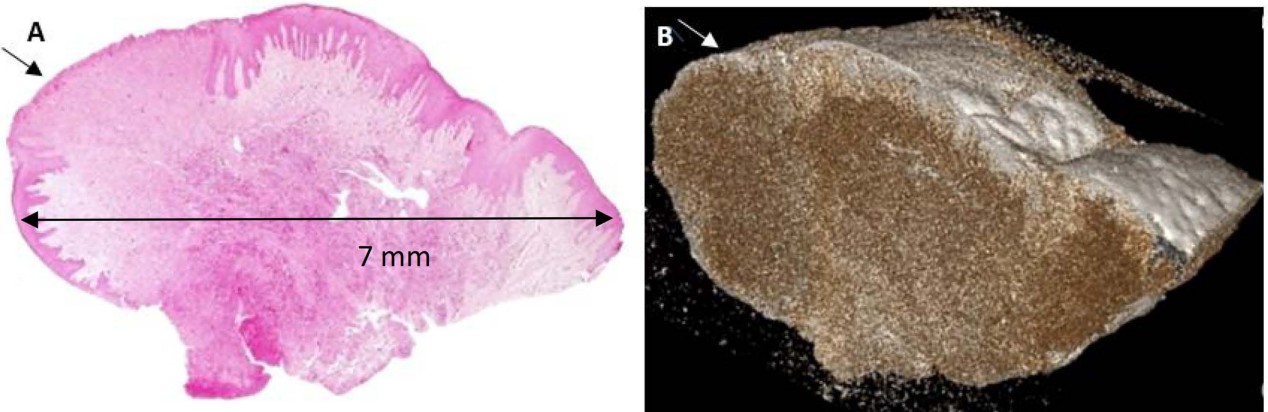

**Fig 3. Fibrous Epulis, ulcerated. A.** Histopathological section stained with H&E. A squamous epithelial lining is observed with an ulcerated area (arrow). **B.** Three-dimensional µCT reconstruction section demonstrating findings consistent with the histopathological section, including the ulcerated area (arrow). Additionally, it provides a high quality and detailed view of the lesion's surface in terms of its irregularities thickness dimension.

centers. Recent advancements propose PTA or nanoparticle-based agents, which could reduce staining to 6–8 hours while improving contrast quality and minimizing tissue distortion, offering a promising avenue for optimization [31]. Additionally, a recent study demonstrated the use of eosin, a common histological dye, for µCT imaging of axillary lymph node biopsies in breast cancer patients, achieving high-contrast 3D visualization of metastatic foci [42]. This approach not only preserved tissue for subsequent histological analysis but also highlighted eosin's potential as a faster and clinically compatible staining alternative, stressing its potential in oral soft tissue applications. The 70 kV, 3 µm scanning protocol was robust for our cohort's small biopsies (0.4–1.0 cm), but denser or thicker lesions, such as those with calcified components or fibrosis, may require higher energy (80–90 kV) or finer resolution (1–2 µm). For instance, chronic inflammatory lesions occasionally exhibit microcalcifications, necessitating adjusted parameters to balance contrast and penetration [43]. The absence of standardized µCT protocols for soft tissues, a persistent challenge noted across literature, underscores the importance of our preliminary protocol as a foundation for broader applications [26].

Integration of µCT into clinical workflows faces logistical barriers, starting from limited scanner availability. µCT scanning significantly exceeds histopathology in terms of its requiring cost-benefit analyses to justify investment in many pathology units, often restricting access to referral centers or research facilities. Clinical integration depends not only on the procurement of µCT devices, but also on long-term planning for sustainability, technical support, and equitable accessibility within the oral pathology service framework. Scanner size and complexity further limit accessibility, as most µCT systems are bulky, research-oriented, and require specialized facilities, maintenance, and trained operators [44]. Addressing the specific domain of oral pathology, its use requires specialized training for dental pathologists and radiologists, and the need for workflows that match histopathology's rapid turnaround (~1–2 days) [44]. Maintenance is equally critical, as routine calibration, software updates, and servicing by specialized engineers are necessary to ensure image quality, safety, and reliability over time. Data storage and processing also present challenges, as high-resolution scans generate large datasets (gigabytes per sample), necessitating robust computational infrastructure [45]. These logistical constraints highlight the need for technological advancements to streamline µCT's integration into clinical use. Given that these technical obstacles will be overcome, a most advantageous future use of the µCT is the examination of metastatic lymph node dissection specimens from oncologic patients. Such a specimen can comprise from a few to tens of H&E-stained slides, yet they provide the pathologist with one section plane of the nodes. However, the metastatic deposits of the malignant tumor can be present in planes not included in the examined H&E-slides and therefore can be easily missed,

compromising the prognosis and survival of the patients. In routine daily practice, H&E sections of all submitted lymph nodes in a multi-planar manner are not feasible, as this alone might completely lock out the workforce of any laboratory. However, μCT scanning of the lymph nodes with the appropriate staining and scanning protocols, will enable the thorough examination of their full volume and cancel or at least minimize, the possibility to miss metastatic deposits, thereby compromising patients' accurate disease staging and adequate treatment.

### Study limitations

The small sample size (n = 12) and focus on benign lesions restrict generalizability to complex or malignant pathologies. Recruiting larger, diverse cohorts, including leukoplakia, dysplasia, and OSCC, would strengthen validation. The single staining protocol (Lugol's, 12 hours) may not suit all lesion types; testing PTA, nanoparticles, or eosin, as demonstrated in recent studies [34], could enhance efficiency and contrast, particularly for vascular or fibrotic tissues [46]. The fixed scanning parameters (70 kV, 3 μm) were effective but may require adjustment for larger or denser specimens, warranting parametric studies to explore 50–90 kV and 1–2 μm resolutions. Manual image analysis, while rigorous, is time-intensive and prone to subjectivity, suggesting future exploration of computational methods, though not implemented here to align with the study's scope [42].

### Conclusions

μCT offers a promising complementary tool for oral soft tissue diagnostics, delivering high-contrast 2D and 3D images that address histopathology's limitations, including malorientation and 2D visualization constraints. Lugol's iodine staining enabled robust visualization of tissue architecture, though cellular detail was inferior to H&E sections. Preliminary protocols for staining and scanning were established, but broader validation with diverse pathologies and optimized workflows is essential for clinical adoption. By enhancing diagnostic accuracy and preserving tissue for further analyses, μCT could transform oral pathology, improving patient outcomes.

### Author contributions

**Conceptualization:** Marilena Vered, Lazar Kats.

**Investigation:** Lazar Kats, Yankel Gabet, Uzi Shpunt.

**Project administration:** Uzi Shpunt.

**Resources:** Uzi Shpunt.

**Validation:** Lazar Kats, Yankel Gabet.

**Visualization:** Marilena Vered, Lazar Kats, Uzi Shpunt.

**Writing – original draft:** Marilena Vered, Lazar Kats.

**Writing – review & editing:** Marilena Vered, Lazar Kats, Yankel Gabet, Uzi Shpunt.

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
