## [Decision Letter · Decision Letter 0]

3 Sep 2025

Dear Dr. Vered,

Thank you for submitting your manuscript to PLOS ONE. After careful consideration, we feel that it has merit but does not fully meet PLOS ONE’s publication criteria as it currently stands. Therefore, we invite you to submit a revised version of the manuscript that addresses the points raised during the review process.

We look forward to receiving your revised manuscript.

Kind regards,

Ronell Bologna-Molina

Academic Editor

PLOS ONE

Journal Requirements:

2. In the online submission form, you indicated that all relevant data are within the manuscript. The full volume of scanned tissues, if needed, can be provided upon request.

3. Please remove all personal information, ensure that the data shared are in accordance with participant consent, and re-upload a fully anonymized data set.

Reviewers' comments:

Reviewer's Responses to Questions

**Comments to the Author**

1. Is the manuscript technically sound, and do the data support the conclusions?

Reviewer #1: Yes

Reviewer #2: Yes

2. Has the statistical analysis been performed appropriately and rigorously?

Reviewer #1: N/A

Reviewer #2: Yes

3. Have the authors made all data underlying the findings in their manuscript fully available?

Reviewer #1: Yes

Reviewer #2: Yes

4. Is the manuscript presented in an intelligible fashion and written in standard English?

Reviewer #1: Yes

Reviewer #2: Yes

Reviewer #1: In addition to the limitations already discussed, the manuscript should address further important challenges for clinical adoption of micro-CT in oral pathology. These include:

(1) cost and infrastructure requirements while integration into routine workflows is mentioned, more detail on equipment acquisition, maintenance, and availability would contextualize feasibility;

(2) need for specialized training interpreting micro-CT images requires specific expertise for both pathologists and radiologists, which may slow uptake; (3) lack of standardized protocols there is currently no universally accepted staining and acquisition protocol for soft tissues, which may affect reproducibility between centers;

(4) data management high-resolution volumetric datasets can be several gigabytes in size, necessitating robust storage and transmission infrastructure; and

(5) risk of contrast-induced artifacts Lugol’s iodine can cause tissue shrinkage or hardening if concentration or staining duration are not optimally controlled.

The study’s design involves imaging and histology on different halves of each biopsy. As a result, the micro-CT and histopathological images are not from identical tissue planes, which limits the ability to make direct, point-by-point comparisons between modalities.

This introduces an inherent topographic mismatch that could explain some of the observed differences seen.

Finally, the authors should state whether any tissue alteration was observed during or after micro-CT scanning, including potential heating effects from microCt high-resolution imaging,

Reviewer #2: maybe it is important explain that micro ct is expensive for this kind of application yet, if you compare with the Histopathological examination process cost. So, it is a good tool but has this disadvantage, too much expensive yet.

**Do you want your identity to be public for this peer review?** For information about this choice, including consent withdrawal, please see our Privacy Policy

Reviewer #1: No

Reviewer #2: **Yes: ** Gainer Raul Jasa Andrade

---

## [Author Response · Author response to Decision Letter 1]

19 Sep 2025

Sept 19th, 2025

Dr. Ronell Bologna-Molina

Academic Editor

PlosOne

Re: Revision of manuscript #PONE-D-25-42054

Dear Dr. Bologna-Molina,

We would like to thank you for reading our submitted manuscript and the opportunity to revise it. We thank the reviewers for their enriching comments.

All comments have been carefully read and discussed, and the manuscript has been revised accordingly. Below are our point-to-point responses. Specification for the changes done in the text relate to the page numbers and lines in the track changes version. We hope that you will find the revised version appropriate for publication in PlosOne.

Sincerely,

Prof. Marilena Vered

Reviewer #1

In addition to the limitations already discussed, the manuscript should address further important challenges for clinical adoption of micro-CT in oral pathology. These include:

(1) cost and infrastructure requirements while integration into routine workflows is mentioned, more detail on equipment acquisition, maintenance, and availability would contextualize feasibility; (2) need for specialized training interpreting micro-CT images requires specific expertise for both pathologists and radiologists, which may slow uptake; (3) lack of standardized protocols there is currently no universally accepted staining and acquisition protocol for soft tissues, which may affect reproducibility between centers;

(4) data management high-resolution volumetric datasets can be several gigabytes in size, necessitating robust storage and transmission infrastructure; and (5) risk of contrast-induced artifacts Lugol’s iodine can cause tissue shrinkage or hardening if concentration or staining duration are not optimally controlled.

Response:

We thank the reviewer for this comment. We have further expanded the original paragraph that addressed µ‎CT limitations, as requested. This can be seen on p. 14, lines 370-374 and pp. 14-15, lines 390-403.

The study’s design involves imaging and histology on different halves of each biopsy. As a result, the micro-CT and histopathological images are not from identical tissue planes, which limits the ability to make direct, point-by-point comparisons between modalities.

This introduces an inherent topographic mismatch that could explain some of the observed differences seen.

Response:

The HE-section and corresponding tomographs are from the same half of tissue. The reviewer is quite right to note that the H&E-stained section and corresponding tomograph might not be from the same plane, however even if not identical they are very similar. This enabled us a good comparison between outcomes of these two methods. This comment was addressed on p. 9, lines 220-224.

Finally, the authors should state whether any tissue alteration was observed during or after micro-CT scanning, including potential heating effects from microCt high-resolution imaging,

Response:

After scanning, tissue was rinsed from Lugol and underwent routine processing and HE staining procedures with an ultimate outcome of sections with no heat-related artefacts or other tissue alterations, as shown in Figs 1A, 2A and 3A. This has been emphasized for each of the presented cases, p. 9, lines 216-218, p. 10, lines 251-242 and p. 11, lines 275-276.

Reviewer #2

Reviewer #2: maybe it is important explain that micro ct is expensive for this kind of application yet, if you compare with the Histopathological examination process cost. So, it is a good tool but has this disadvantage, too much expensive yet.

Response:

We thank the reviewer for this comment. This was addressed on p. 14, lines 392-396.

---

## [Decision Letter · Decision Letter 1]

15 Oct 2025

Micro-Computed Tomography as a complementary tool for histopathological diagnosis of oral soft tissue lesions – Proof of concept

PONE-D-25-42054R1

Dear Dr. Vered,

We’re pleased to inform you that your manuscript has been judged scientifically suitable for publication and will be formally accepted for publication once it meets all outstanding technical requirements.

Kind regards,

Ronell Bologna-Molina

Academic Editor

PLOS ONE

Additional Editor Comments (optional):

Reviewers' comments:

Reviewer's Responses to Questions

**Comments to the Author**

Reviewer #2: All comments have been addressed

2. Is the manuscript technically sound, and do the data support the conclusions?

Reviewer #2: Yes

3. Has the statistical analysis been performed appropriately and rigorously?

Reviewer #2: Yes

4. Have the authors made all data underlying the findings in their manuscript fully available?

Reviewer #2: Yes

5. Is the manuscript presented in an intelligible fashion and written in standard English?

Reviewer #2: Yes

Reviewer #2: (No Response)

**Do you want your identity to be public for this peer review?** For information about this choice, including consent withdrawal, please see our Privacy Policy

Reviewer #2: **Yes: ** Gainer Jasa

---

## [Editor Report · Acceptance letter]

PONE-D-25-42054R1

PLOS ONE

Dear Dr. Vered,

I'm pleased to inform you that your manuscript has been deemed suitable for publication in PLOS ONE. Congratulations! Your manuscript is now being handed over to our production team.

Kind regards,

on behalf of

Professor Ronell Bologna-Molina

Academic Editor

PLOS ONE